# Reactive Oxygen Species and the Redox-Regulatory Network in Cold Stress Acclimation

**DOI:** 10.3390/antiox7110169

**Published:** 2018-11-21

**Authors:** Anna Dreyer, Karl-Josef Dietz

**Affiliations:** Department of Biochemistry and Physiology of Plants, Faculty of Biology, University of Bielefeld, 33615 Bielefeld, Germany; andreyer@techfak.uni-bielefeld.de

**Keywords:** chilling stress, cold temperature, posttranslational modification, regulation, ROS, thiol redox network, thioredoxin

## Abstract

Cold temperatures restrict plant growth, geographical extension of plant species, and agricultural practices. This review deals with cold stress above freezing temperatures often defined as chilling stress. It focuses on the redox regulatory network of the cell under cold temperature conditions. Reactive oxygen species (ROS) function as the final electron sink in this network which consists of redox input elements, transmitters, targets, and sensors. Following an introduction to the critical network components which include nicotinamide adenine dinucleotide phosphate (NADPH)-dependent thioredoxin reductases, thioredoxins, and peroxiredoxins, typical laboratory experiments for cold stress investigations will be described. Short term transcriptome and metabolome analyses allow for dissecting the early responses of network components and complement the vast data sets dealing with changes in the antioxidant system and ROS. This review gives examples of how such information may be integrated to advance our knowledge on the response and function of the redox regulatory network in cold stress acclimation. It will be exemplarily shown that targeting the redox network might be beneficial and supportive to improve cold stress acclimation and plant yield in cold climate.

## 1. Plant Response to Cold Temperature

Biological systems are unavoidably affected by changes in ambient temperature. Such temperature effects particularly concern the temperature-dependent rates of spontaneous and enzyme-catalyzed chemical and physical reactions, the structural and molecular dynamics, and strength of molecular interactions. Each of these effects interferes with the state of metabolism and cellular signal processing. Many plants have evolved the capacity to adapt to low-temperature climates and develop locally distinct adaptive traits [1]. Thus, the response of photosynthetic metabolism to 4 °C varies not just between different species, but also between differently adapted populations. Oakley et al. [1] demonstrated this phenomenon with Arabidopsis from Italy and Sweden, and their crossing, by measuring the fast recovery of non-photochemical quenching after 2 min of darkening. The cold-acclimation response involves a profound reorganization of gene expression and posttranscriptional processes employing abscisic acid (ABA)-independent and dependent pathways [2]. The fraction of unsaturated fatty acid residues in cell membranes in particular linoleic acid (C18:2) increases at the expense of saturated lipids during chilling stress acclimation [3]. The fatty acid desaturation increases the fluidity of the membrane at a lower temperature and involves activation of fatty acid desaturases such as stearoyl-acyl carrier-protein desaturase (SAD) [4]. In addition, higher amounts and often different types of sugars and other osmolytes accumulate at low temperatures. Osmolyte accumulation is assumed to prepare for dehydration stress during freezing-induced water deficit [5]. In addition, the antioxidant defense is enhanced during cold stress acclimation [6,7]. In concert with reactive oxygen species (ROS) production, the antioxidant system controls the redox regulatory network of the cell. Our review focuses on this aspect of chilling stress acclimation and is intended to synthesize a broader perspective on redox network function and consequences.

## 2. Cold Stress Experiments in the Laboratory 

Chilling stress occurs if non-primed plants or non-adapted plants, which are unable to be hardened, are exposed to cold temperatures significantly below growth temperature. Table 1 summarizes eight experiments of recent years. The genetic model plants *Arabidopsis thaliana, Oryza sativa,* and *Sorghum bicolor*, the genetic model *Capsella bursa-pastoris*, two *Jatropha* species with potential in biofuel production, and the medicinal plant *Calendula officinalis* were grown at 20–28 °C during the light phase and 18–28 °C during the night and then mostly transferred to 4 °C, or 10, 12, and 15 °C, the latter for *Capsella*, rice, and soybean. Thus, the chosen temperature regimes covered down-shifts between 14 and 24 °C and were scrutinized for 30 min [8] to 14 d [6]. It is noteworthy that recovery from cold stress attracts increasing attention in *Arabidopsis thaliana* [6]. These authors recognized a positive relationship between the speed of recovery from 14 d chilling treatment and the strength of the plastid antioxidant system [6]. Thus, the *Arabidopsis* accessions N14, N13, Ms-0, and Kas-1, which grow under quite low temperatures in nature, displayed a weaker expression of some plastid antioxidant genes during the phase of 14 d cold treatment and simultaneously maintained a primed state after transfer to normal temperature. Maintenance of a primed state has an advantage if periods of chilling stress will return, but is disadvantageous for growth. Thus, maintenance of primed state has a trade-off for fitness at elevated temperatures [6].

## 3. Central Role of the Redox Regulatory Network in Stress Acclimation

All cells express a regulatory network of thiol-containing proteins which integrates information from reductive metabolism and electron drainage from the network by ROS [14]. Targets of regulation by thiol oxidation are metabolic enzymes of various pathways, signal transduction elements, transcription and translation factors, and, thus, essentially all functional levels of the cell [15]. These target proteins are reduced by the concerted action of redox input elements and redox transmitters.

Redox input elements transfer electrons from metabolism to the redox transmitters. The nicotinamide adenine dinucleotide phosphate (NADPH)-dependent thioredoxin reductase A (NTRA) functions as such redox input element in the cytosol by transferring electrons from NADPH to thioredoxin (TRX). The same role is played by NTRB in the mitochondrion [16] and by NTRC in the chloroplast and other plastids [17]. NTRC is a variant which carries a TRX domain in addition to the NTR domain. An additional input element is the ferredoxin-dependent thioredoxin reductase (FTR) which transfers electrons from the photosynthetic electron transport chain (ETC) to TRX [18].

Decisive elements in redox regulation are TRX which constitutes large families in plants comprising proper TRX and TRX-like proteins. They function as redox transmitters. A genome-wide association study (GWAS) with soybean accessions grown at 28 °C and transferred to 15 °C for cold stress (Table 1) [13] discovered 143 genomic sites considered as promising for improving cold acclimation of soybean. Soybean is a C4 plant which is chilling sensitive. The study focused on photosynthetic performance during cold treatment and recovery. Among the identified genes of interest were two TRX genes, *Sb03g004670* and *Sb06g029490*, which may contribute to the cold acclimation variation among the accessions [13]. These TRX-genes link cold stress to the regulation of the Calvin cycle [13,19,20].

Thiol peroxidases have a very high affinity to H_2_O_2_, alkylhydroperoxides and peroxynitrite (ONOO^-^). This group of enzymes comprises peroxiredoxins (PRX) and glutathione peroxidase-like proteins (GPX) and was suggested to control the spreading of peroxide signals in the cell. They act as redox sensors [21,22]. Their affinity and abundance support their function as primary reactants with peroxides. Recently, it was shown that the chloroplast 2-cysteine peroxiredoxin functions as thioredoxin oxidase and thereby co-controls the activation state of target proteins of redox regulation, such as malate dehydrogenase, phosphoribulokinase, and fructose-1,6-bisphosphatase [23]. Based on this study on chloroplast 2-cysteine peroxiredoxin it may be hypothesized that other thiol peroxidases play the same role in the redox regulatory network of plastids and other cell compartments.

The final electron acceptors of the network are peroxides which oxidize the redox sensors. Generator systems are the photosynthetic and respiratory ETC, substrate oxidases in peroxisomes and apoplast, and the NADPH oxidases in the plasma membrane [24]. Antioxidants counteract the accumulation of ROS and, thus, lower the electron drainage from the network (Figure 1). Oxidation of TRX by O_2_, direct oxidation of target proteins by peroxides or other mechanisms likely contribute to the oxidation of network components. But these mechanisms are poorly understood and may lack the specificity which is needed for tailored responses.

## 4. Variability of Cold Response Between Species

If the ambient temperatures drop significantly below growth temperature, an imbalance between photosynthetic light and dark reactions is established, and the photosynthetic electron transport chain releases more superoxide (O_2_^−^) by electron transfer to oxygen as alternative electron acceptor. An O_2_^−^ increase is observed during chilling stress, e.g., in *Cynodon dactylon* [25]. 

Several studies indicate that increased superoxide dismutase (SOD) activity likely is related to chilling tolerance. SOD catalyzes the formation of hydrogen peroxide (H_2_O_2_) and H_2_O from two superoxide molecules and two protons [26,27]. The response of SOD varies among species. Plants contain several *SOD* genes which often show species- and stress-specific expression patterns [28]. However, activity measurements usually cannot easily discriminate among the isoforms of Mn-, CuZn- and Fe-SOD. The transcript amounts and activity of SOD increase during low-temperature treatment in *Capsella bursa pastoris L.* [9] and *Calendula officinalis* [10]. The SOD activity in *Calendula officinalis* rises less than the transcript amount pointing to posttranscriptional regulation of synthesis or higher protein degradation rates [10,29]. Likewise, the SOD activities of *Phaseolus vulgaris* [30], *Cynodon dactylon* [25], and *Withania somnifera* [31] only increase slightly upon exposure to low temperature. In the ecotype AGB025 of *W. somnifera* the SOD activity was even constant during the first three days of low-temperature treatment. In a converse manner, the ecotype AGB002, which is known to be more tolerant to chilling stress than AGB025, showed an increased enzyme activity [31]. 

Similar contrasting results were observed for two cell cultures derived from *O. sativa* subspecies *Japonica cv.* Nipponbare and *Indica cv.* 9311. The genotype *Japonica* known to be more adapted to chilling temperatures showed induced SOD activity after 24 h of low-temperature treatment, whereas the activity in *Indica* was constant during the 72 h of stress treatment [32,33]. The significant role of SOD is plausible if considering the role of ROS in addressing the thiol network as discussed above. H_2_O_2_ acts as an efficient electron sink while O_2_^-^ does not, but rather may cause damage, e.g., by peroxidation of lipids.

Unlike SOD, the catalase (CAT) activity in *O. sativa Indica* increased during the first 24 h of cold treatment but later on, returned to control level. CATs are localized in the peroxisomes and decompose H_2_O_2_ released from peroxisomal oxidases like glycolate oxidase with its role in photorespiration. CAT activity is important to minimize leakage of H_2_O_2_ from the peroxisome to the cytosol. The constitutively high SOD activity together with the transiently induced activity of CAT, could tune the accumulation of H_2_O_2_. In contrast to *Indica, Japonica* rice showed a relatively high, but constant, activity of CAT [32]. A similar increase of SOD and CAT activity was observed *C. bursa-pastoris* [9], *Citrullus lanatus* [34], and *C. dactylon* [25]. Therefore, SOD and CAT activity could play an important part during chilling stress, although some species like *C. officinalis* [10] and *P. vulgaris* [30] fail to increase the activity of these enzymes. The increased CAT is likely important to avoid accumulation of H_2_O_2_ if its production increases under cold due to imbalances and changes in metabolism.

The ascorbate-dependent Foyer–Halliwell–Asada cycle reduces H_2_O_2_ to water and is located in several subcellular compartments. This pathway relies on ascorbate peroxidases (APX) [27]. APX catalyzes the reduction of H_2_O_2_ to water by using ascorbate as electron donor, which results in the formation of dehydroascorbate (DHA). The produced DHA is recycled to ascorbate by dehydroascorbate reductase (DHAR) using glutathione as reductant (GSH). Liberated oxidized glutathione (GSSG) is regenerated by glutathione reductase (GR) using NADPH [26]. An ascorbate-independent water-water cycle for H_2_O_2_ reduction employs GPX and PRX which are found in plastids, mitochondria, and cytosol. The thiol peroxidases are part of the thiol network as described above. The oxidized thiol peroxidase reacts with an electron donor like TRX or glutathione with or without glutaredoxin (GRX) [35,36]. 

The APX activity decreased in *C. bursa pastoris* during cold treatment of 1 to 5 days duration. This is quite unexpected in the light of high transcript levels observed during the first 24 h of low-temperature treatment [9] and could be due to posttranscriptional regulation [37,38] or APX inactivation by reaction with H_2_O_2_ in the absence of ascorbate [39]. APX inactivation could be a means to enable H_2_O_2_ accumulation for signaling. Quantification of APX protein levels would help to explain the weak correlation between transcript level and enzyme activity. To compensate for the instability of enzymes and to increase the antioxidant capacity, a constant supply of *de novo* synthesized proteins is needed [40]. This may explain why the transcript levels of the corresponding antioxidant enzymes usually increase under stress, e.g., in *C. officinalis* where both the transcript level and the activity of APX increased with prolonged time of low-temperature treatment [10]. APX activity also increased in cold-tolerant *J. macrocarpa*, whereas the APX activity decreased >6-fold in the cold-sensitive *J. curcas* [11]. The ascorbate-dependent water-water cycle is supported by stimulated ascorbate biosynthesis, strengthened glutathione homeostasis, enhanced sulphate reduction, and TRX pathway [41]. 

Apparently, different species reveal partly controversial responses of components of the antioxidant system. Considering the central role of redox homeostasis in cell function, this variation is surprising. Thus, the dependency of priming and resilience of the primed state on the strength of the chloroplast antioxidant system offers an important explanation for this variation [6].

## 5. The Compartment-Specific Response of the Components of the Redox-Network to Cold in *A. thaliana*

The enzymes of the redox-regulatory network are localized in different subcellular compartments of the cell (Figure 2A,C). Frequently more than one isoform of the redox elements is present in the different compartments. The three schemes assemble published transcript changes for the three subcellular compartments cytosol, chloroplast, and mitochondrion during a time course of 24 h cold treatment [8].

As outlined above, NADPH-dependent thioredoxin reductases (NTR) function as redox input elements and isoforms are present in each compartment. The relative transcript amounts of the cytosolic *NTRA* (Figure 2A), chloroplast *NTRC* (Figure 2B) and mitochondrial *NTRB* (Figure 2C) slightly decreased during the 24 h of low-temperature treatment. NTRC, which regenerates oxidized 2-Cys-PRX in an NADPH dependent manner [17,73], showed a similar decrease in expression as *2-Cys-PRXB* (Figure 2B). The redox transmitters TRX and GRX transfer reducing equivalents to target proteins such as thiol peroxidases [74]. The downregulation of the h-type TRXs in the cytosol occurred in a peculiar time-dependent manner. *TRX-h3* was downregulated during the first 30 min, *TRX-h9* after 1 h, followed by *TRX-h4*, *TRX-h2* and *TRX-h1* which only changed after 24 h of low-temperature treatment (Figure 2A). 

Transcript levels of most of the redox sensors *GPX* and *PRX* were also downregulated during the low-temperature treatment. Small upregulation of cytosolic and mitochondrial GPX6 was an exception. In comparison to the downregulated transcript level of the PRXs after short-term cold stress, the transcript level of *2-Cys-PRXA*, *PRXQ,* and *PRXIIE* increased after 14 days of cold stress treatment [6]. Transcripts for other enzymatic antioxidants like *APX*, *monodehydroascorbate reductase (MDAR), dehydroascorbate reductase (DHAR)* and *SOD* slightly decreased during the low-temperature treatment, whereas an increase of transcript level was observed for *sAPX* and *MDAR* after 14 days of chilling stress [6]. Although the *APX* and *SOD* transcript levels decreased during the first 24 h of cold treatment, the activity of both enzymes increased [75]. Later on after 4–8 days, APX and SOD activity reached the activity of untreated plants. This could be due to posttranscriptional regulation [76]. It would be interesting to scrutinize the weak relationship of transcript levels and enzyme activity by quantification of APX and SOD protein levels. 

ROS are continuously generated in the photosynthetic and respiratory electron transport chain and during photorespiration [77,78]. Previously, it was estimated that about 30% of the electrons of the photosynthetic electron transport chain flow into ROS metabolism [79,80]. These figures were recently challenged by Driever and Baker [81] who only detected low amounts of ROS generated by the photosynthetic ETC. The authors proposed that this ROS releases feature signalling functions rather than serving as a major alternative electron acceptor. Furthermore, ROS are generated under normal conditions at low rates by NADPH oxidases (RBOH) [82]. The transcript amounts of *RBOH* were downregulated, tentatively suggesting that the formation of ROS decreases at low temperatures. However, the activity and the protein level of these enzymes will have to be analyzed, since transcript levels poorly reflect the abundance or activity of their gene product [77]. 

In summary, it is striking that the main changes in transcript level only occurred after 24 h of low-temperature exposure. There were some exceptions: *APX2* is considered as a sensitive marker of oxidative stress and is upregulated under conditions of excess excitation of the photosynthetic apparatus [83]. Here, *APX2* was upregulated 2.4-fold after 30 min and maximally 5.6-fold after 6 h of cold treatment. Juszczak et al. [80] detected profound transcript changes after 14 d of cold treatment showing the persistence of metabolic disturbance at low temperatures. Thus, a global transcriptome analysis at later time points would be interesting to extend our analysis.

Furthermore, the cytosolic redox-regulatory network was stronger regulated than that of the chloroplast and mitochondria. The relatively small changes in transcript levels for chloroplast proteins of the network could be due to the setup in this particular experiment, where the light intensity was decreased from 150 µmol photons m^−2^ s^−1^ during growth to just 60 µmol photons m^−2^ s^−1^ during the low-temperature treatment [8]. The lowering of the incident photosynthetic active radiation probably minimizes the development of photoinhibition [84]. 

*A. thaliana* Col0 is sensitive to chilling stress and displays stunted growth, reduced metabolism and low biomass accumulation at low temperature [75]. Therefore, the transcriptional downregulation of the redox-regulatory network could participate in the chilling sensitivity of *A. thaliana* Col0.

## 6. Improvement of Cold Tolerance by Modulating the Redox-Network

Manipulation of the plant redox-network can enhance cold-stress tolerance [75,85,86,87]. Moon et al. [87] showed that overexpression of NTRC under control of the cauliflower mosaic 35S (CaMV35) promoter in *A. thaliana* enhances the tolerance to cold stress and freezing. These authors discussed NTRC and that it might affect the formation of secondary structures of mRNA at low temperatures to stimulate translation or protect nucleic acids against oxidative stress [87]. Since NTRC functions as a redox input element, it would be interesting to study the impact of NTRC overexpression on the redox network of the plastid including 2-Cys-PRX [17,82]. *2-Cys-PRXB,* and *NTRC* showed similar decreases in transcript levels during short-term chilling stress (Figure 2B).

Improved cold tolerance was observed in tomato (*Lycopersicon esculentum*) overexpressing tAPX [85]. The tAPX overexpressing lines also revealed disturbed GSH/GSSG ratios which could be due to the impact of chloroplast redox state on GSH biosynthesis [88]. Furthermore, the high tAPX level in the transgenic tomatoes resulted in a decreased photoinhibition of photosystem I and II under chilling stress [85]. Increased NTRC and tAPX activity will shift the redox state of the chloroplast to a more reduced state, either by stimulated feeding of electrons into the network (NTRC) or decreasing the electron drainage to ROS. 

Shafi et al. [75] transferred both Cu/Zn-SOD from *Potentilla atrosanguinea* and APX from *Rheum australe* (with high similarity to the peroxisomal *APX* gene of *A. thaliana*) under control of the 35SCaMV promoter into *A. thaliana*. Since *P. atrosanguinea* and *R. australe* grow in alpine climate, the authors hypothesized that their enzymatic system might be better adapted to stress than *A. thaliana*. Total soluble sugars and proline content increased in the transgenic line, which could help to maintain the membrane integrity. As a result, the electrolyte leakage was decreased under cold treatment at 4 °C compared to the wildtype (WT) [75]. The accumulation of cytosolic O_2_^●−^ and H_2_O_2_ should be suppressed in these plants and, indeed, less ROS accumulated in the transgenic lines.

The overexpression of the cytosolic redox transmitter AtGRXS17 in tomato resulted in an enhanced chilling tolerance [86]. The activities of SOD, CAT, and heme peroxidases were higher in the AtGRXS17 overexpressing lines compared to the WT, while the transcript levels were almost identical. These results indicate that the enhanced chilling tolerance mediated by AtGRXS17 is associated with the activity and/or stability, rather than the transcript level, of the enzymes. Other reasons for the enhanced chilling tolerance could be due to the increase in total soluble sugars, the reduced accumulation of H_2_O_2_, and the reduced electrolyte leakage [85]. This particular GRX was previously shown to link redox and ROS homeostasis with auxin signaling and development [89].

These findings demonstrate that improved cold tolerance can be engineered in plants by manipulating elements of the redox-regulatory network and underline the significance of the network in chilling stress acclimation. 

## 7. Conclusions

The state of the thiol redox regulatory network is under the control of electron input by metabolism and electron drainage to ROS. ROS amounts, in turn, are controlled by the activity of ROS generator systems and antioxidant defense systems. Conclusive evidence shows that redox network elements control important features of the cold acclimation and recovery program. This review focused on the thiol-disulfide network and its interaction with ROS and the antioxidant defense system. However, other players, such as classical hormones including abscisic acid, salicylic acid, and oxylipins, interfere with acclimation to chilling temperatures. More recently novel players were discovered to affect cold acclimation. They include γ-amino butyric acid and melatonin [90]. Keeping this in mind, a few conclusions can be drawn for future directions.
(1)Cold stress acclimation experiments often focus on leaves and photosynthetic metabolism. Response heterogeneity of different cell types has scarcely been addressed. Cell type-specific transcriptome, proteome, and metabolome analyses should reveal how other cell types respond to cold stress. But these approaches remain challenging and laborious. (2)Only a few methods allow researchers to address subcellular compartments. Transcriptome data provide easy access due to the predicted and often proven subcellular localization of the encoded gene products. This approach is straightforward and was applied here to the redox regulatory network. It would be interesting to see this type of data processing more frequently. However, the transcript amount is poorly linked to protein amount and activity. For a full understanding, we need compartment-specific proteomics and enzyme activity tests.(3)Metabolite-profiling of non-aqueous tissue fractions is another method which provides access to the major subcellular compartments. Non-aqueous fractions reflect the metabolic state of the compartments *in vivo* and are obtained from previously frozen and freeze-dried plant material like leaves [91]. This method was recently applied to cold-stress *A. thaliana* [92]. The latter study did not include metabolites with direct significance in the redox regulatory network.(4)Subcellular and cellular specificity can be addressed by imaging technologies detecting specific physicochemical properties such as Ca^2+^-activity, specific compounds or the redox state of the glutathione system by using roGFP coupled to GRX [93]. The roGFP:GRX sensor can be targeted to different cell compartments and should be used to explore the glutathione redox state in dependence on cold stress intensity and duration.(5)To describe the state of the redox network in subcellular compartments, mathematical modeling and simulation combined with redox-proteomics for validation will be required. A pioneering modeling study presented a simulation of the fluxes through the ascorbate-dependent water-water cycle [94] and most recently, the thioredoxin oxidase-dependent inactivation of chloroplast enzymes was simulated [23]. Conceptually, cold stress appears to be an interesting target for this kind of simulation and prediction.(6)The question of acclimation and damage during the cold period is certainly of significant interest. However, the costs of priming and the speed of recovery likely play a major role when it comes to fitness and competitiveness. Thus, the report by Juszczak et al. [6] deserves attention as it provides clues on the advantages and disadvantages of expressing a strong antioxidant system.

## Figures and Tables

**Figure 1 antioxidants-07-00169-f001:**
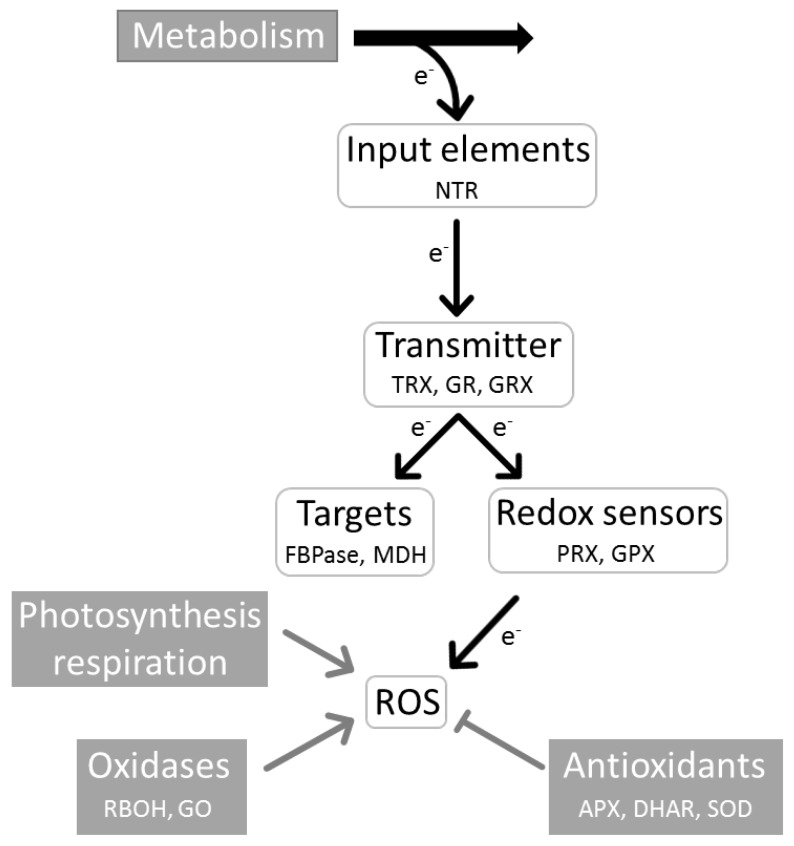
Basic structure of the redox regulatory network of the plant cell. Dependent on its state, metabolism feeds regulatory electrons via input elements (NTR: Nicotinamide adenine dinucleotide phosphate (NADPH)-dependent thioredoxin reductase) into the network. Redox transmitters (TRX: thioredoxin; GR: glutathione reductase, GRX: glutaredoxin) transfer electrons to regulated target proteins (FBPase: Fructose-1,6-bisphosphatase, MDH: malate dehydrogenase). Redox sensors (GPX: glutathione peroxidase (like), PRX: peroxiredoxin) also drain electrons from the transmitters in dependence on the ROS amount. The ROS amount is controlled by the activity of generator systems such as photosynthetic and respiratory electron transport chains and oxidases (RBOH: NADPH oxidase; GO: glycolate oxidase) and the decomposition of reactive oxygen species (ROS) by the antioxidant system (APX: ascorbate peroxidase, DHAR: dehydroascorbate reductase, SOD: superoxide dismutase). The proteins mentioned in the figure are typical representatives.

**Figure 2 antioxidants-07-00169-f002:**
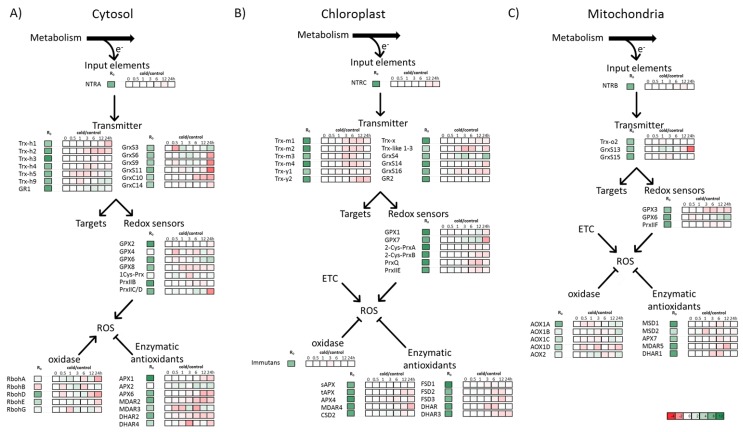
Time-dependent change in transcript levels of the redox regulatory network in the cytosol (**A**), chloroplast (**B**), and mitochondrion (**C**) during chilling stress. Relative transcript levels in *A. thaliana* were obtained from the Gene Expression Omnibus (GEO) Database of National Center for Biotechnology Information (NCBI) (Accession GSE5620 for control conditions and GSE5621 for low temperature treated plants) [8]. R0 shows the transcription level normalized by the Affymetrix system. The other boxes indicate the transcript levels in cold treated plants compared to the control conditions at the depicted time point. A complete list with the precise log2-fold changes is provided in the supplement. AOX: Alternative oxidase; APX: Ascorbate peroxidase; CSD: Cu/Zn-superoxide dismutase; DHAR: Dehydroascorbate reductase; ETC: electron transport chain; FSD: Fe-superoxide dismutase; GR: Glutathione reductase; GRX: Glutaredoxin; GPX: Glutathione peroxidase; MDAR: Monodehydroascorbate reductase; NTR: NADPH-dependent thioredoxin reductase; PRX: Peroxiredoxin; Rboh: Respiratory burst oxidase homolog protein; TRX: Thioredoxin. The assignment of subcellular localization of the proteins listed in this figure was based on the references [42,43,44,45,46,47,48,49,50,51,52,53,54,55,56,57,58,59,60,61,62,63,64,65,66,67,68,69,70,71,72], see also Appendix A.

**Table 1 antioxidants-07-00169-t001:** Experimental parameters used to explore chilling stress acclimation. See text for further description of the experiments.

Species	Plant Age	Growth Condition	Cold Treatment		Other Comments	Reference
			duration	temperature		
*Arabidopsis thaliana*	16 days	16/8 h light/dark 24 °C	0; 0.5; 1; 3; 6; 12; 24 h	4 °C	Decreased light intensity during cold stress treatment	[8]
*Arabidopsis thaliana*	42 days	16/8 h light/dark 20 °C/18 °C	14 days	4 °C	Decreased light intensity during cold stress Additional 1,2, and 3 days of deacclimation	[6]
*Capsella bursa pastoris L.*	30 days	16/8 h light/dark 25 °C	24; 48; 72; 96; 120 h	10 °C		[9]
*Calendula officinalis*	14 days	16/8 h light/dark 25 ± 2 °C	24; 48; 72; 96; 120 h	4 °C		[10]
*Jatropha curcas*	45 days	14/10 h light/dark 28 °C	48 h	4 °C	Partially pretreated at 15 °C for five days Cold sensitive *Jatropha*	[11]
*Jatropha macrocarpa*	45 days	14/10 h light/dark 28 °C	48 h	4 °C	Partially pretreated at 15 °C for 5 days, cold tolerant *Jatropha*	[11]
*Oryza sativa*	14 days	14/10 h light/dark 28 °C/22 °C	6 days	12 °C	2 days pretreatment with melatonin	[12]
*Sorghum bicolor*	30 days	16/8 h light/dark 28 °C/24 °C	6 days	15 °C	5 days recovery	[13]

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
