# Peer review of "Reactive Oxygen Species and the Redox-Regulatory Network in Cold Stress Acclimation"

_antioxidants, 2018, doi:10.3390/antiox7110169_

Round 1
Reviewer 1 Report
The manuscript is well prepared and written.
Line 68, it should be "ROS", instead of "reactive oxygen species".
Line 78, it should be "constitutes".
Line 118, it should be Italic (gene name).
Line 271, it should be "straightforward".
Author Response
The manuscript is well prepared and written.
· Line 68, it should be "ROS", instead of "reactive oxygen species". done
· Line 78, it should be "constitutes". done
· Line 118, it should be Italic (gene name). done
· Line 271, it should be "straightforward". done
Reviewer 2 Report
This review provides important insight into the redox regulatory networks that are associated with reactive oxygen species (ROS) and cold stress acclimation in plants. The authors synthesize a variety of studies on this topic and draw a few conclusions for future directions to advance the field. This review is highly appropriate for the journal Antioxidants.
Below are a few minor comments to improve the quality of the manuscript for publication:
Ln 35-37 The content of this sentence is somewhat narrow in the context of the introduction. Please provide the reader with some type of transition to Arabidopsis or broaden the scope of this sentence.
Ln 38, define ABA upon first use in text
Ln 44 the enhancement of antioxidant defense during cold stress acclimation should be referenced
Ln 57 please indicate which model organism was used to demonstrate a correlation between the speed of recovery from chilling treatment and the strength of the antioxidant system
Ln 58 The authors tend to provide a little too much detail in certain sections of the review before defining what those details mean - e.g., Arabidopsis accessions N14, N13, Ms-O and Kas-1? Do a reader who is not working with Arabidopsis, this type of distinction is unclear in terms of Arabidopsis biology etc.
Table 1 - would help to provide a brief (one phrase) summary of the overall findings for each experimental study
Ln 68 Targets (add s)
Ln 85 provide reference to support linkage of the Trx-genes to the regulation of the Calvin cycle
Ln 86-88 provide reference(s) for the first two sentences of this paragraph
Ln 99 incomplete sentence
Ln 101 define the type of cell that is meant by [the cell] e.g., plant cell
Ln 118 sod genes - italics? be sure to use proper nomenclature throughout
Ln 116-126 the reader is left wondering what the overall conclusion of this paragraph is until they reach ln 131-133 - recommend reorganizing this section a bit
Ln 134-135 which type of rice? Japonica? Please clarify in the text
Ln 136 how is Indica related to the discussion? Please clarify in the text
Ln 140 please clarify at this point how the increase in SOD and/or CAT relates to cold stress tolerance
Ln 140-142 this info would be helpful to be placed earlier in paragraph for the reader to first be introduced to catalase
Ln 151 define APX in text upon first use and explain relevance of APX to reader regarding redox balance, ROS production etc. early in this section.
Ln 194 italics proofread
Ln 217-221 need to add a least one citation to this paragraph
Ln 237-238 recommend avoiding single sentence paragraphs - need to synthesize the info a bit more
Ln 244 define WT upon first use
Ln 249 activity and/or stability (not activity and stability) - since do not yet have evidence for both levels
Ln 285-286 incomplete sentence
Figure 2 - too detailed for this type of review - recommend drawing a cartoon figure to represent the general trends that were observed by the transcriptomic studies
Author Response
This review provides important insight into the redox regulatory networks that are associated with reactive oxygen species (ROS) and cold stress acclimation in plants. The authors synthesize a variety of studies on this topic and draw a few conclusions for future directions to advance the field. This review is highly appropriate for the journal Antioxidants.
Below are a few minor comments to improve the quality of the manuscript for publication:
1. Ln 35-37 The content of this sentence is somewhat narrow in the context of the introduction. Please provide the reader with some type of transition to Arabidopsis or broaden the scope of this sentence. done
2. Ln 38, define ABA upon first use in text done
3. Ln 44 the enhancement of antioxidant defense during cold stress acclimation should be referenced. done
4. Ln 57 please indicate which model organism was used to demonstrate a correlation between the speed of recovery from chilling treatment and the strength of the antioxidant system. done
5. Ln 58 The authors tend to provide a little too much detail in certain sections of the review before defining what those details mean - e.g., Arabidopsis accessions N14, N13, Ms-O and Kas-1? Do a reader who is not working with Arabidopsis, this type of distinction is unclear in terms of Arabidopsis biology etc.. done by simplifying
6. Table 1 - would help to provide a brief (one phrase) summary of the overall findings for each experimental study. Most of the experiments listed in the Table 1 are discussed in the text, we thought that it is not necessary to give a one sentence summary to each experiment in the legend of the table.
7. Ln 68 Targets (add s). done
8. Ln 85 provide reference to support linkage of the Trx-genes to the regulation of the Calvin cycle. done
9. Ln 86-88 provide reference(s) for the first two sentences of this paragraph. The reference is listed in the next sentence. The first and second sentence are connected.
10. Ln 99 incomplete sentence. done
11. Ln 101 define the type of cell that is meant by [the cell] e.g., plant cell. done
12. Ln 118 sod genes - italics? be sure to use proper nomenclature throughout. done
13. Ln 116-126 the reader is left wondering what the overall conclusion of this paragraph is until they reach ln 131-133 - recommend reorganizing this section a bit. For improvement we now start this section with: Several studies indicate that increased SOD activity likely is related to chilling tolerance. This gives the reader the needed orientation.
14. Ln 134-135 which type of rice? Japonica? Please clarify in the text. done
15. Ln 136 how is Indica related to the discussion? Please clarify in the text. done
16. Ln 140 please clarify at this point how the increase in SOD and/or CAT relates to cold stress tolerance. Do deal with this point, we have added the following sentence: The increased CAT likely is important to avoid accumulation of H2O2 if its production increases under cold due to metabolic imbalances and changes.
17. Ln 140-142 this info would be helpful to be placed earlier in paragraph for the reader to first be introduced to catalase. To address this point, we have moved the introductory sentence to catalase function to the beginning (second sentence) of the paragraph.
18. Ln 151 define APX in text upon first use and explain relevance of APX to reader regarding redox balance, ROS production etc. early in this section. done
19. Ln 194 italics proofread. done
20. Ln 217-221 need to add a least one citation to this paragraph. done
21. Ln 237-238 recommend avoiding single sentence paragraphs - need to synthesize the info a bit more. We have combined paragraphs at two places to deal with this point.
22. Ln 244 define WT upon first use. done
23. Ln 249 activity and/or stability (not activity and stability) - since do not yet have evidence for both levels. done
24. Ln 285-286 incomplete sentence. done
25. Figure 2 - too detailed for this type of review - recommend drawing a cartoon figure to represent the general trends that were observed by the transcriptomic studies. We kindly ask to keep this figure in its present format. The reason is that the reader should get some awareness for the complexity. We refer to other review like that of Mittler (TIPS, 2004) which is much more detailed and exactly for this reason became important references for many other authors.